# Minigene as a Novel Regulatory Element in Toxin-Antitoxin Systems

**DOI:** 10.3390/ijms222413389

**Published:** 2021-12-13

**Authors:** Barbara Kędzierska, Katarzyna Potrykus

**Affiliations:** Department of Bacterial Molecular Genetics, Faculty of Biology, University of Gdańsk, Wita Stwosza 59, 80-308 Gdańsk, Poland; katarzyna.potrykus@ug.edu.pl

**Keywords:** minigene, mini-ORF, translation regulation, gene expression, toxin-antitoxin system, *axe-txe*

## Abstract

The *axe-txe* type II toxin-antitoxin (TA) system is characterized by a complex and multilayered mode of gene expression regulation. Precise and tight control of this process is crucial to keep the toxin in an appropriate balance with the cognate antitoxin until its activation is needed for the cell. In this report, we provide evidence that a minigene encoded within the *axe-txe* operon influences translation of the Txe toxin. This is the first example to date of such a regulatory mechanism identified in the TA modules. Here, in a series of genetic studies, we employed translational reporter gene fusions to establish the molecular basis of this phenomenon. Our results show that translation of the two-codon mini-ORF displays an *in cis* mode of action, and positively affects the expression of *txe*, possibly by increasing its mRNA stability through protection from an endonuclease attack. Moreover, we established that the reading frame in which the two cistrons are encoded, as well as the distance between them, are critical parameters that affect the level of such regulation. In addition, by searching for two-codon ORFs we found sequences of several potential minigenes in the leader sequences of several other toxins belonging to the type II TA family. These findings suggest that this type of gene regulation may not only apply for the *axe-txe* cassette, but could be more widespread among other TA systems.

## 1. Introduction

Toxin-antitoxin (TA) systems are widely present in bacterial and archaeal genomes, as well as on mobile genetic elements, such as phages and plasmids, where they are often found in multiple copies [1,2,3]. They are usually composed of two genes, encoding a toxin which can be regarded as a kind of molecular bomb targeting one of the crucial cellular processes, and a cognate antitoxin which protects the cell against the toxin’s attack [4,5]. Thus, it is vital for the microorganism to keep a proper balance between these two elements. Currently, seven types of TA systems have been proposed on the basis of the antitoxin’s nature and its mode of action [6,7,8]. The most abundant and best-studied is type II, where both components function as proteins. A proper stoichiometry of the toxin and its antidote, which is nontoxic for the cell, is typically achieved at the transcription initiation step, when the two genes are co-expressed from a common promoter that is negatively autoregulated by the toxin-antitoxin protein complex. However, there are known examples of TA cassettes that are regulated in a diverse, often more intricate, manner [4,5,6,9].

One such module is *axe-txe* derived from the pRUM plasmid of an *Enterococcus faecium* strain [10]. In our previous work, we have shown that regulation of this cassette is complex and multilayered, when examined in the *Escherichia coli* cells. Apart from the main *p_at_* promoter that drives expression of the antitoxin and toxin genes, we have identified some other regulatory elements, such as internal promoters within the *axe* and *txe* genes, an additional promoter upstream of *p_at_*, and a hairpin structure at the 3′ end of the *txe* transcript, as illustrated in Figure 1 [11,12]. The *p_axe_* promoter found within the *axe* gene was shown to direct the synthesis of a transcript encoding the *txe* toxin [11]. As this promoter is roughly located in the middle of the *axe* gene, its transcript must possess a 114 nt-long leader fragment upstream of the start codon of the *txe* toxin (Figure 1). Closer inspection of this region’s nucleotide sequence revealed the existence of three potential start codons—two ATG and one GTG—all of which were in frame with the same TAA stop codon. All of these codons were also in the same reading frame with the *txe* ATG start codon. These findings prompted here to analyze if any of these potential minigenes might somehow affect expression of the downstream *txe* gene, making the regulation within the whole *axe-txe* operon even more complex than previously described.

Several recent reports on transcriptome analysis, ribosome profiling, and bioinformatic predictions demonstrated that small open reading frames (ORFs) are abundant in bacterial genomes, including 5′ untranslated regions (5′-UTR) of many genes [13,14,15,16,17]. However, to date, only single examples of mini-ORFs in bacterial biology have been studied. Translation of an upstream ORF (uORF) can have positive or negative consequences on the expression of the downstream gene, and can modulate its expression in different ways. The classic example is the tryptophan operon of *E. coli*, where translation of a 14-amino-acid open reading frame (*trpL*) within the leader affects transcription of the operon’s main genes by enforcing different secondary structures in mRNA, forming a terminator or anti-terminator, depending on the availability of tryptophan [18]. A similar transcription-attenuation mechanism has been described for several other amino acid synthesis operons, and for some other cistrons, including *pyrBL* of *E. coli* and *mgtA* of *Salmonella enterica*, implicated in pyrimidine biosynthesis and the transport of magnesium, respectively [19,20]. The uORFs can also directly influence translation of the downstream gene, as has been shown for several antibiotic-resistant genes. In such instances, the drug molecule binds to the ribosome’s exit tunnel and directly interacts with the nascent peptide. Such immediate interaction induces ribosome stalling, and was demonstrated to free the ribosome-binding site (*rbs*, also known as the Shine–Dalgarno sequence) for the downstream gene via changes in mRNA secondary structure. This kind of regulation is in place, for example, for the *cat* and *erm* antibiotic-resistant genes [21].

Recently, translation of small ORFs which overlap with a downstream gene was found to play regulatory roles on the latter. The translation of *pssL* and *yoaL* exerted positive effects on expression of the main ORFs, *pssA* and *yoeE*, respectively. On the other hand, the translation of *baxL* and *argL* minigenes exerted negative influence on the *baxA* and *argF* cistrons’ expression levels, respectively [15]. Two mini-ORFs have been also identified in the *Bacillus subtilis ycbK* translation initiation region [22]. These dipeptide-encoding genes are located 5 and 10 nucleotides upstream of the *ycbK* gene, and all share the same Shine–Dalgarno (SD) sequence. Expression of these minigenes was shown to negatively interfere with translation of the downstream gene [22]. A six-codon minigene was also found in the long and AU-rich leader of the *LEE4* operon in enterohemorrhagic *Escherichia coli* (EHEC) [23]. This mini-ORF is preceded by a strong *rbs*. It has been shown that occupancy of this SD element by ribosomes protects *espADB* mRNA from degradation by RNase E, through hindering its access to the AU-rich cleavage sites. In this way, expression of the downstream genes is positively regulated at the mRNA stability level [23].

Nevertheless, a direct inspiration for studies presented here was a paper from Prof. S.J.W. Busby’s lab, where a minigene identified in the leader fragment of the EHEC *LEE1* transcript was shown to influence the activity of the downstream *ler* gene, which encodes a transcription factor which is important for the expression of pathogenicity determinants [24]. The inactivation of this two-codon minigene resulted in the inability of bacteria to properly interact with epithelial cells. Moreover, the authors demonstrated that the distance between the mini-ORF and the main cistron determines whether the effect is stimulatory or inhibitory [24].

In this work, we present a series of genetic studies providing evidence that a minigene located within the leader region of the *p_axe_* transcript enhances the translation efficiency of the downstream-located *txe* toxin gene. This is the first example to date of a mini-ORF modulating expression of a toxin belonging to a toxin-antitoxin system. Our study provides further evidence on the abundance of diverse regulatory mechanisms ensuring the properly balanced expression of TA cassettes.

## 2. Results

### 2.1. A Two-Codon Minigene Located within the Leader of p_axe_ Transcript Has a Positive Effect on the Txe Gene Translation

We began our investigation on potential minigenes in the *p_axe_* mRNA leader sequence with the construction of appropriate translational fusions in the pRW225 vector, kindly provided by Prof. S.J.W. Busby. This low-copy-number RK2-based and tetracycline-resistant plasmid allowed us to clone DNA fragments in front of the *lacZ* reporter gene deprived of its own translational signals [24]. Our first aim was to assess the translation potential of all start codons identified in the *p_axe_* transcript leader region. To do this, we cloned DNA fragments composed of the *p_axe_* promoter, and extended the sequence to each of these codons. In this way, the translation initiation signals encompassed in the cloned sequences were fused in frame with the *lacZ* reporter gene (Figure 2A). The resulting recombinant plasmids were transformed into a ∆5∆lac *E. coli* strain, a derivative of SC301467, in which we performed deletion of the entire lactose operon by P1 transduction. This host was chosen to avoid any possible cross-interactions, since it carries deletions of 5 out of 13 known type II chromosomal TA cassettes, including *yefM-yoeB* (a homolog of *axe-txe*), ref [25]. β-galactosidase activity was measured according to the procedure developed by Miller [26]. The results, which are presented in Figure 2B, show that neither ATG1 nor GTG provide translational start signals to drive the synthesis of the reporter protein. On the other hand, the ATG2 and ATGtxe fusions drive significant production of β-galactosidase, 1100 MU, and 594 MU, respectively, meaning that both of them are associated with functional translational start signals. This is in agreement with the fact that the ATG1 and GTG fusions do not possess a properly located Shine–Dalgarno sequence, while the other two start codons are preceded by a seemingly functional ribosome-binding site. The SD sequence is composed in most cases of 4–6 nucleotides, which correspond to the longer UAAGGAGGU mRNA fragment, complementary to the anti-SD sequence in 16S rRNA. GGAG, GAGG, and AGGA are regarded as strong core *rbs*, while AAGG is a weaker variant. The aligned spacing between the start codon and SD, counting from its center (underlined G), was established to be effective in the range of 7 to 15 nucleotides [27,28]. It was also shown that the AU-rich spacers increase translation efficiency and allow for more flexibility in the SD-AUG spacing [27,29]. Thus, our results argue that the ATG2 fusion possesses functional translation signals, and this translation must terminate at the TAA stop codon, meaning that the two-codon minigene ATGactTAA, which we called *mg2*, undergoes the translation process.

Since the stop codon for *mg2* and the start codon for *txe* are separated by 15 nt, i.e., 5 codons, we decided to investigate a possible effect of this minigene on the *txe* gene translation. For this purpose, we created a series of pRW_ATGtxe-derived constructs, in which we introduced mutations in order to inactivate its subsequent elements, as is illustrated in Figure 3A. These recombinant plasmids were transformed into a ∆5∆lac *E. coli* strain to monitor β-galactosidase activity. When ATG1 was changed into ACG (fusion ATGtxe_ATG1mut), a 21% reduction in β-galactosidase activity was observed in comparison to the wild-type fragment. The GTG codon overlaps with *rbs* for *mg2*, so a mutation which changes GTG into CTG also weakens this Shine–Dalgarno sequence (aagGTG to aagCTG) (fusion ATGtxe_GTG/SDmut). However, this nucleotide change did not make any substantial difference in the reporter protein activity when compared to the wild-type construct (only a 10% decrease). Next, we introduced mutations in *mg2*, first changing only the ATG, and then both the ATG and TAA stop codons. Disruption of the ATG codon only (changed into ACG, fusion ATGtxe_ATG2mut) reduced the reporter gene activity by 34%, while an 83% decrease in β-galactosidase activity was observed when both the start and stop codon for *mg2* were mutated: ATG to ACG, and TAA to TTA (fusion ATGtxe_mg2mut, Figure 3). These results clearly indicate that translation of *mg2* positively influences expression of the downstream gene, and when it is inactivated, β-galactosidase activity decreases by about 7 folds. However, the 13 codon-long *mg1,* which starts at ATG1 and ends at the same TAA stop codon as *mg2*, slightly increases this effect, since some further decrease in β-galactosidase activity was observed when ATG1, ATG2, and TAA were inactivated (fusion ATGtxe_mg1,2mut, 88% decrease). Thus, assuming that the changes in the reporter protein activity reflect the *txe* toxin expression, we have identified a novel mode of gene expression regulation in the *axe-txe* toxin-antitoxin system.

Next, we asked what would happen if we deleted the whole leader region containing minigenes. Thus, we constructed a pRW_ATGtxe_w/o_mg1,2 fusion which was deprived of the whole DNA sequence with minigenes, as indicated in Figure 3A and Appendix A. In comparison to the wild-type fusion-containing minigenes (ATGtxe), this new construct deprived of the minigenes’ sequences (ATGtxe w/o mg1,2), produced 30% more β-galactosidase activity, meaning that the level of *txe* expression would be significantly increased (Figure 3B).

### 2.2. The mg2 Minigene Acts in Cis to Regulate Translation of Txe and Does Not Display Toxicity towards E. coli Cells

In order to assess whether the *txe* leader minigenes perform their activity *in cis* or *in trans*, we transformed the ∆5∆lac *E. coli* strain with pRW_ATGtxe or pRW_ATGtxe_mg2mut, along with pBAD33 derivatives, in which a DNA fragment containing *mg1*,*2* or just *mg2* was placed under control of the arabinose-inducible *p_BAD_* promoter. L-arabinose was added to induce the minigenes’ expression, and the reporter protein production in these cells was monitored, as described above. The results, depicted in Figure 4, indicate that there is no statistically significant difference in β-galactosidase activity upon supplying the minigenes *in trans*, meaning that the *mg2mut* phenotype is not complemented by a plasmid-carrying inducible wild-type *mg2* supplied *in trans*. If these minigenes worked *in trans*, an increase in *lacZ* activity should be observed in cells carrying the LacZ reporter fused to the *txe* leader with mutated *mg2* (the pRW_ATGtxe_mg2mut plasmid), in comparison to cells where the expression of these minigenes, driven from *p_BAD,_* was repressed by the addition of glucose. Indeed, such an upregulation was not observed.

Expression of very short ORFs may cause the inhibition of the translation process and the arrest of bacterial cell growth [30]. Toxicity occurs because peptidyl-tRNA hydrolase fails to recycle the peptidyl-tRNA released from ribosomes at the stop codon of a minigene rapidly enough. This causes depletion in tRNA corresponding to the last sense codon in the mini-ORF. It was demonstrated that the toxicity level depends on different conditions, including the length of a minigene, the last sense codon, and *rbs* strength [31,32]. To examine if products of the minigenes encoded in the *txe* leader are toxic, we used the above-described pBAD33_mg1,2 and pBAD33_mg2 plasmids, in which minigene induction occurs upon addition of L-arabinose. Bacterial growth was monitored every 60 min for 5 h. We did not notice any significant differences in the optical density between bacterial cultures that grew in the presence of 0.2% L-arabinose, which stimulates *mg2* or *mg1*,*2* induction, in comparison to cells carrying empty pBAD33 (Appendix A). Thus, we can conclude that the minigenes present in the *txe* leader do not affect *E. coli* cells’ growth.

### 2.3. The Distance between mg2 and Txe, as Well as the Reading Frame of Both Genes, Are Important Features Modulating Expression of the Latter

Knowing that *mg2* presents an *in cis* mode of action, we attempted to determine whether the mini-ORF and the gene located downstream must be in the same open reading frame and whether the precise distance between these two cistrons is important for the regulation of the *txe* translation process. In order to achieve this, we constructed a series of pRW225_ATGtxe derivatives, in which the distance between the stop codon of the minigene and the SD sequence of *txe/lacZ* was successively decreased or increased by one nucleotide. It was possible to decrease this distance by a maximum of 2 nucleotides so as to not disrupt the Shine–Dalgarno sequence of *txe*. On the other hand, we increased this distance by a maximum of 6 nucleotides, i.e., by 2 codons (Figure 5A).

The results depicted in Figure 5B indicate that it is not necessary for the minigene to be in the same reading frame as the downstream gene to modulate its expression. As can be seen, the highest activity of the reporter protein is observed in the case of ATGtxe+1 construct, where both start codons are in different reading frames. Interestingly, the minigene present in each of the three reading frames can influence expression of the downstream gene, but to a different extent. On the other hand, the distance between the two genes (*mg2* and *txe*/*lacZ*) seems to be a more important parameter. It appears that there is an optimal distance between the two genes yielding the highest activity in each frame, and when it is increased or decreased by one codon, the minigene’s amplifying effect is reduced (Figure 5). However, this correlation seems to hold true only for constructs that maintained the original reading frame, or where it was shifted to the +1 frame (green and yellow bars in Figure 5B, respectively). For some reason, this effect was not observed for the constructs in the +2 frame (−1, +2 and +5; blue bars in Figure 5B).

In parallel, as a control, we tested similar constructs where *mg2* was inactivated by mutations in its start and stop codons (Figure 6A). Surprisingly, the activity of some of the reporter fusions remained at a high level (Figure 6B). A closer look at the DNA sequences revealed that in the −1, +2 and +5 variants, all of which are in the same reading frame, two additional stop codons (TGA), now in frame with ATG1 and GTG, have been created (Figure 6A). This raised a possibility that upon *mg2* inactivation, ATG1 or GTG perhaps take over, and one of them begins to function as a start codon for a new minigene whose translation ends at these newly created stop codons. To test this hypothesis, we created double minigene mutants, ATGtxe_ATG1mut_mg2mut and ATGtxe_GTGmut_mg2mut, both with the −1 and +2 variants. Measurements of β-galactosidase activity show that their translation is significantly reduced when both, *mg1* and *mg2*, are mutated, while GTG inactivation in the absence of *mg2* does not cause a decrease in reporter gene activity (Figure 7).

These results confirm our assumption and indicate that when *mg2* is not functional, ATG1, but not GTG, starts to serve as the initiation codon for a newly created minigene. The new mini-ORF is 15 and 16 codons long, when −1 and +2 variants are employed, respectively. In both cases, the first stop codon is located 5 nt upstream of ATGtxe, while the second stop codon overlaps by 1 nt with ATGtxe. Coming back to the results presented in Figure 5, it appears that this newly created minigene may act even in the presence of a functional *mg2* (Figure 5, blue bars), and this is why the nucleotide deletions or insertions which were introduced in order to change the distance between the *mg2* stop and *txe* start, did not make a difference in reporter gene activity.

Taken together, our results show that both the reading frame and the distance between the minigene and the downstream cistron have an influence on the level of translation of the latter. Based on the Txe regulation example, we can also see that when nucleotide deletion or insertion creates a new minigene—despite the fact that it is longer and its stop codon is closer to the ATGtxe—it is able to regulate Txe translation in a similar way to the wild-type construct. Thus, these results argue that the length of a minigene, at the range tested in this study, is not critical for its functioning as a translational regulator of the downstream gene. Rather, it seems that the distance from the minigene’s stop codon to the start codon of the second gene is a crucial parameter.

### 2.4. Two-Codon Mini-ORFs Are Present within the Antitoxin Gene of the relBE and mazEF TA Families in Diverse Bacterial Species

In order to assess if the two-codon minigenes exist in any other TA cassettes, we searched for two-codon ORFs with the ATG start codon and TAA, TGA, or TAG stop codons within other antitoxins’ DNA sequences. The TA nucleotide sequences were obtained from the Toxin-Antitoxin Data Base (TABD2.0) [33]. We thus probed 140 sequences of antitoxins from the *yefM-yoeB*, *relBE*, *mazEF*, and *phd-doc* TA families (Appendix A). We found 15 potential minigenes in *relBE* and 2 in *mazEF* family TA cassettes, as presented in Table 1. With the exception of the two examples indicated in the table by asterisks, the rest of the identified minigene were encoded in the same reading frame as the toxin gene. We also analyzed the last sense codon in terms of the encoded amino acid. There does not seem to be any preference for a particular type of amino acid. Moreover, we also compared the distance between the ATG of the minigene and the ATG of the toxin. In most cases, this distance was much longer than in the *axe-txe* system, and ranged between 3–174 nucleotides (Table 1). In addition, we identified potential Shine–Dalgarno sequences upstream of all these minigenes. Finally, we also identified sequences within the antitoxin genes which could potentially play a role of internal promoters (data not shown).

However, it must be noted that our search was concerned with only two-codon minigenes, which does not exclude the existence of longer mini-ORFs within the antitoxin genes that can affect a given toxin’s expression.

## 3. Discussion

Here, we report for the first time the engagement of a mini-ORF in translation regulation of a toxin belonging to a toxin-antitoxin system. The *axe-txe* system is a type II TA module which was first identified on the multidrug-resistant pRUM plasmid in a clinical isolate of *Enterococcus faecium* [10]. Our previous studies demonstrated that this cassette is regulated in a very complex and multilayered manner, when present on a plasmid introduced in *E. coli* cells [11,12]. We are using this model bacterium as a TA element-bearing plasmid host, since the expression vectors suitable for enterococci that are currently available are characterized by low detection sensitivity, and thus enable only the strongest elements to be studied (our unpublished data). In this study, we used a genetic approach to demonstrate that the two-codon minigene located within the leader region of *txe* influences the translation efficiency of the toxin gene (Figure 1).

The *p_axe_* promoter is an internal promoter identified within the *axe* antitoxin gene, which drives expression of an additional portion of the Txe toxin [11]. Within *txe’s* 114 nucleotide-long leader sequence, we have identified three potential translation start sites: ATG1, GTG, and ATG2—all of which are in the same reading frame with the *txe* methionine start codon. There is also a TAA stop codon in the same reading frame which terminates translation of these potential mini-ORFs (Figure 1). First, we showed here that ATG2, when fused to the *lacZ* reporter gene deprived of its own translation signals, possesses a strong translation ability, in contrast to ATG1 and GTG codons. In comparison, the construct bearing the *p_axe_* promoter with the whole leader up to the *txe* ATG fused to the reporter gene (ATGtxe construct), showed a substantial but lower translation activity (Figure 2). Our mutational analysis shows that *mg2* plays a major role here; however, *mg1* also contributes to the effect mediated on *txe* expression. When we inactivated both the *mg1* and *mg2* minigenes, the activity of the construct was decreased by over 8 folds; however, when the entire fragment with minigenes was deleted (ATGtxe_w/o mg1,2), the translational ability of the ATG *txe* increased by 30%, in comparison to the ATGtxe construct (Figure 3). This means that the presence of minigenes upstream of *txe/lacZ* increases translation efficiency, while the complete lack of this sequence makes *txe* translation even more effective.

These results may appear inconsistent; however, they indicate a possible explanation of the minigene regulation. The *p_axe_*-derived leader sequence is 65% made up of adenosines and uridines (74 out of 114 nucleotides), with two evident AU-rich clusters, including one (UUUAAAA) located between the SD and ATG for *mg2*, and the other—AAAAAAUU, upstream of SD for *mg1*. It has been shown that such sequences could be targets, for example, for RNase E which cleaves RNA at AU-rich regions without defined sequence specificity, or for ribosomal S1 protein, an essential component of the *E. coli* translation machinery, responsible for mRNA selection and the enhancement of translation efficiency [34,35,36,37]. Similar RNA-binding specificity for S1 and RNase E should not be surprising since this endonuclease possesses the so-called S1 domain (responsible for binding) [35]. Thus, it is possible that some kind of competition between the ribosome and the ribonuclease may occur at the *txe* leader sequence. It is not unreasonable to suspect that the ribosome’s binding to *rbs* of the minigenes may protect the mRNA from ribonuclease degradation, increasing its stability and thereby also its translation efficiency. When there is only a short leader with the SD sequence for *txe* translation (without minigenes and AU-rich sequences), the reporter’s gene activity is high (Figure 3—ATGtxe_w/o mg1,2 construct). In case of the wild-type leader sequence, the efficiency of β-galactosidase activity decreases; however, it is still kept at a high level due to the hypothesized ribosome protection of the AU-rich regions next to *mg1* and *mg2* (ATGtxe construct). On the other hand, when *mg1* and *mg2* are inactivated (ATGtxe_mg1,2mut construct) the AU-rich regions are unprotected and nuclease can access them, thus decreasing the stability of the transcript and lowering protein quantity.

The above-hypothesized mechanism would be similar to the one proposed by Lodato and co-workers for *esp* mRNA in the EHEC *LEE4* operon, where a ribosome bound to a six-codon mini-ORF protected this transcript from the RNase E attack [23]. The authors showed that neither the mini-ORF translation nor the hexapeptide itself regulate translation of the downstream gene, but effective binding to the strong Shine–Dalgarno sequence upstream of the minigene is sufficient for the observed stabilization effect. These authors also suggested the same solution for the regulation of the *ler* gene in the EHEC *LEE1* operon, where a two-codon minigene has been shown to positively influence translation of the main ORF [23,24]. However, our data indicate that active translation of the minigene is needed for increased reporter protein/Txe activity, since mutations in both start and stop codons—without disrupting the Shine–Dalgarno sequences—caused a decrease in *txe* translation (Figure 3). Our results seem to confirm the assumption that the role of minigenes might be to protect mRNA against an attack from a ribonuclease, and in this way, they increase the rate of *txe* translation. This hypothesis is further corroborated by observations that frequently translated mRNAs are efficiently protected from nucleolytic degradation [38]. As a matter of fact, some authors indicate that mRNA may be stabilized simply by an efficient ribosome loading at the *rbs* [23,39,40], while others persuade that only actively translating ribosomes can protect mRNA from rapid decay, and mRNA stability is dependent on the level of its translation [36]. Regardless of certain discrepancies in the precise mechanism of this process, the final result is the same and lies in mRNA protection against nucleolytic degradation.

We also wanted to assess if the potential minigene we identified displays an *in trans* or *in cis* mode of action, and if its overproduction influences bacterial growth. The reason for that was the observation that overexpression of some minigenes may be toxic for bacterial cells [30]. The data gained from studies with a synthetic library of two-codon minigenes indicated that this toxicity depends on the identity of the second (and final) sense codon; different degrees of toxicity result from different rates of peptidyl-tRNA released from the ribosome [41]. Moreover, it was also verified that the minigene toxicity is inversely correlated with the length of the coding sequence, indicating two-codon mini-ORFs as the most toxic [31]. In addition, when the SD sequence upstream of a minigene is strong, the ribosome reinitiates without dissociating after peptide release, and this increases toxicity [32]. Finally, minigenes ending at the UAA codon were shown to be more toxic in comparison to those with UGA and UAG stop codons; this results from faster drop-off and slower termination rates [32].

The *mg2* minigene described here fulfils the above characteristics for a toxic minigene. It is two-codon, its last codon is ACT (encodes for threonine) that has been classified as a toxic codon [41,42], and its stop codon is UAA. However, experiments where we used L-arabinose-inducible *mg2* do not demonstrate the inhibition of bacterial cell growth (Appendix A). Nevertheless, despite the fact that the *txe* leader minigene does not act globally, it may still exert local effects. It has been shown that a long ribosome pause on the last sense codon, arising from inefficient translation termination, eventually results in mRNA stabilization [32,43]. Thus, mRNA stability, mediated by minigenes described previously as toxic, is higher because their transcripts are engaged and protected for a longer time by ribosomes [41]. This elegantly fits our hypothesis that the minigene identified here acts by protecting *p_axe_*-derived mRNA from nuclease degradation. Moreover, our results unambiguously confirm that the potential mini-ORFs located in the *txe* leader sequence work *in cis* since their overproduction from the arabinose-inducible promoter did not complement the *mg2mut* phenotype (Figure 4). This is in agreement with other reports describing minigenes in the leader sequences as *in cis* working elements [22,23,24]. Altogether, these data indicate that the minigenes act locally (*in cis*)*,* by increasing *txe* translation efficiency.

Having established that the *mg2* minigene works *in cis,* we asked if it is important that a mini-ORF and the downstream gene are in the same reading frame, and if the distance between them is crucial for proper functioning. Our data suggest that a minigene can play its assigned role in any of the three reading frames in respect to the next cistron, albeit with different degrees of efficacy. We found that the proper distance between the two cistrons is a more crucial parameter, and when it is changed by one codon the minigene’s amplifying effect is reduced. We suspect that this is related to a translation-coupling mechanism in which two genes undergo translation by the same ribosomal complex if the distance between them is appropriately small. RNase protection experiments and sequence alignment studies indicate that the ribosome covers around 34–38 nt of mRNA. It recognizes the *rbs* sequence in the translation initiation region, and extends the contact on both sides of the initiator codon: about 20 nt towards 5′ and 13 nt towards 3′ ends [44]. If the distance between two cistrons is smaller than the region covered by one ribosome, translational coupling occurs. It can be imagined that the ribosome sitting on SD *mg2* in the wild-type leader covers the SD sequence of *txe*; however, when the distance between these two genes is extended by nucleotide insertions, the *txe*’s *rbs* is no longer in contact with the ribosome sitting on *mg2*. This probably results in the gradual loss of translation coupling ability, and the effect of *mg2* on *txe* expression is eventually lost (Figure 5). In a situation when the distance between the two cistrons is too large for translation coupling but not large enough to accommodate two separate ribosome complexes, translation inhibition of the downstream gene may be observed. The coupled genes’ translation is sensitive to the stop and start codons’ spacing, and varies inversely with the distance between the termination and initiation codons of the two genes. Thus, correct juxtaposition of the stop and start signals is critical for reinitiation, with the best results found with partially overlapping codons [45]. This favors readthrough translation over reinitiation at the first cistron start site, and increases the translation level of the downstream gene. Such an arrangement is present in our constructs with +2 frame translation, where new stop codons are created, one of which overlaps the start of *txe/lacZ* (Figure 5 and 6, blue bars).

The upstream gene in translation coupling may also positively contribute to translation of the downstream gene by locally increasing the concentration of ribosomes, thus increasing ribosome recruitment, or by disrupting unfavorable secondary structures in the translation initiation region of mRNA [46]. Long mRNA leaders are especially susceptible to adopt different secondary structures which can have regulatory effects on expression of the downstream genes. Thus, secondary structures may, for example, control the accessibility of Shine–Dalgarno and start codon sequences for ribosomes. However, the *p_axe_*-derived leader is 65% AU-rich, and when assessed by mFOLD, no evident strong secondary structures were found (Appendix A). Thus, the regulatory effects of the secondary structures potentially created in this region are not expected to influence *txe* expression.

To conclude, we propose that besides mRNA protection, translation coupling is an additional mechanism by which the minigene in the *txe* leader may act. In this aspect, regulation described here would differ from that found in the *LEE1* and *LEE4* operons, since the mini-ORFs there are located too far from the main gene for translation coupling [23,24]. In addition, we believe this is the first reported example of a minigene found to regulate gene expression in the toxin-antitoxin systems. Our search of two-codon mini-ORFs in other systems belonging to different TA families shows that this type of regulation may be more widespread among different bacterial species (Table 1). Further studies at the level of mRNA and direct protein production are needed to verify the hypotheses presented here.

## 4. Materials and Methods

### 4.1. Strains, Plasmids and Oligonucleotides

*E. coli* K-12 DH5α was used for plasmid construction and purification. Strain SC301467, a derivative of MG1655 devoid of *mazF*, *chpB*, *relBE*, *dinJ-yafQ* and *yefM-yoeB* [25] was also devoid of the entire *lac* operon by P1 transduction procedure. The resultant strain, named here ∆5∆lac, carries a kanamycin-resistant cassette. This strain was used for β-galactosidase assays with appropriate derivatives of the pRW225 plasmid [24]. The pBAD33 plasmid was used to clone DNA fragments containing minigenes under L-arabinose-inducible promoter [47]. Bacteria were grown in Luria–Bertani (LB) medium at 37 °C with shaking. Tetracycline and chloramphenicol were added to final concentrations of 12.5 µg/mL and 34 µg/mL, respectively, when required. LB plates were supplemented with X-gal, as a β-galactosidase activity indicator. Plasmids and oligonucleotides used are listed in Appendix A, respectively. Oligonucleotides were ordered from Sigma-Aldrich/Merck (Darmstadt, Germany) or Eurofins Genomics (Ebersberg, Germany) and all restriction enzymes were purchased from Fermentas/ThermoFisher Scientific (Waltham, MA, USA). All plasmid constructs were verified by sequencing (Macrogen Europe).

### 4.2. Cloning and Mutagenesis Procedures

For amplification of the wild-type inserts that were later cloned into pRW225 and pBAD33 vectors, standard PCR technique was used. To introduce different single nucleotide substitutions to pRW_ATGtxe and its derivatives, we applied circular mutagenesis, also known as the QuickChange method (Agillent). This strategy utilizes two overlapping primers carrying desired mutations, which are employed to amplify the whole plasmid in a long-run PCR. It has been essentially performed according to the detailed protocol described in [48]. Because the pRW225 vector is too large (15 kb) for this method, the mutations were introduced separately into the smaller pTE103 plasmid (2.3 kb) carrying the whole *axe-txe* operon (pTE_pat_axe-txe). Next, such mutated derivatives were used as templates in a standard PCR to make pRW225 recombinants. To obtain pRW_ATGtxe_mg2mut where both the start and the stop codons of mg2 are changed, it was necessary to perform Overlap Extension PCR (OE-PCR). Briefly, OE-PCR utilizes four primers (two primers complementary to each other with desired mutations and two outer primers) in two separate PCR reactions, then both products (having short overlapping fragments) are mixed and used as a template for the third PCR run [49]. To obtain pRW_ATGtxe or pRW_ATGtxe_mg2mut derivatives with nucleotide deletions or insertions, we ran standard PCR, in which one of the primers carried the desired mutations. Phusion High-Fidelity DNA Polymerase (ThermoFisher Scientific) and primers listed in Appendix A were used in all these procedures.

### 4.3. Promoter Fusion Studies and β-Galactosidase Assays

Competent cells of ∆5∆lac strain were transformed with derivatives of pRW225 vector bearing the *lacZ* gene under transcriptional control of the *p_axe_* promoter fragments. PCR fragments were cloned into pRW225 between *EcoR*I and *Hind*III restriction sites upstream of promoterless *lacZ* gene. Overnight cultures carrying recombinant plasmids were diluted (1:100) into fresh LB medium and grown with shaking until OD_600_~0.5. Then, β-galactosidase activity assay was performed with cells permeabilized with chloroform and SDS, as described by Miller [26].

### 4.4. Bacterial Growth Rates

Overnight cultures of ∆5∆lac strain carrying pBAD33 or its derivatives—pBADmg2 and pBADmg1,2 plasmids, were diluted (1:100) into fresh LB medium containing 0.2% L-arabinose and grown with shaking at 37 °C. Samples of culture were withdrawn at time intervals, OD_600_ was measured, and the growth curves were obtained.

### 4.5. Bioinformatic Analyses

An in-house computer program was used to search for two-codon minigenes in the sequences of diverse type II toxin-antitoxin systems. Nucleotide sequences of antitoxins from *yefM-yoeB*, *relBE*, *phd-doc* and *mazEF* TA families were downloaded from the Toxin-Antitoxin Data Base (TABD2.0) available on https://bioinfo-mml.sjtu.edu.cn/TADB2/index.php [33]. Secondary structure of *txe* leader fragment was created in mFOLD program (http://mfold.rna.albany.edu/) (accessed on 30 June 2020).

### 4.6. Statistical Analysis

Statistical analyses were performed with the online Mann–Whitney U test calculator (Statistics Kingdom, 2017), available from: http://www.statskingdom.com/170median_mann_whitney.html (accessed on 30 June 2020 ). Student’s two-tailed *t*-test was employed for Figure 3 and Figure 4, while one-way Anova with post hoc Tukey test was used to evaluate data presented in Figure 5 and Figure 6. Differences were deemed as statistically significant for *p* < 0.05.

## Figures and Tables

**Figure 1 ijms-22-13389-f001:**
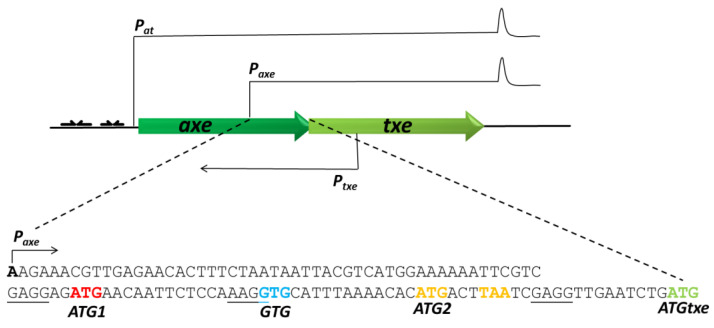
Schematic organization of the *axe-txe* operon with the nucleotide sequence of the leader transcript produced from the *p_axe_* promoter. The *axe* and *txe* genes are depicted by green arrows. Transcripts originating from the *p_at_*, *p_axe_,* and *p_txe_* promoters are indicated by thin black lines and arrows. A hairpin structure is present at the end of the *p_at_* and *p_axe_* transcripts. The leader region of the *p_axe_* transcript starts at the *p_axe_* transcription start site (black A in bold), while it ends at the ATG codon of the *txe* gene (ATGtxe) (in green). Within this leader, additional translation start sites (ATG1, GTG, and ATG2) are marked in different colors, along with the stop codon TAA. Potential Shine–Dalgarno sequences are underlined.

**Figure 2 ijms-22-13389-f002:**
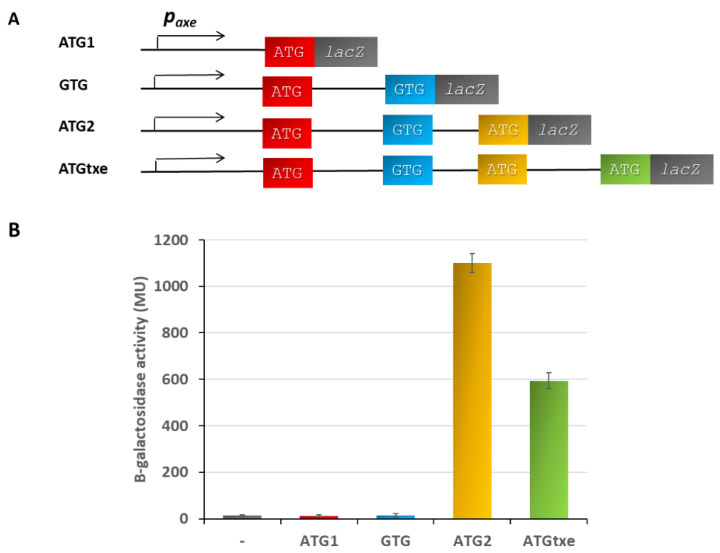
Translation ability of start codons from *p_axe_* leader. (**A**) Schematic organization of the translation fusions where successive ATG and GTG codons were fused in frame with the *lacZ* gene deprived of its own translation signals. (**B**) Appropriate pRW225 vector fusions were introduced into ∆5∆lac *E. coli* strain, and β-galactosidase activity was assessed when cell cultures reached OD_600_ ≈ 0.5. These results are the average of at least three independent experiments; error bars represent standard deviation (S.D.).

**Figure 3 ijms-22-13389-f003:**
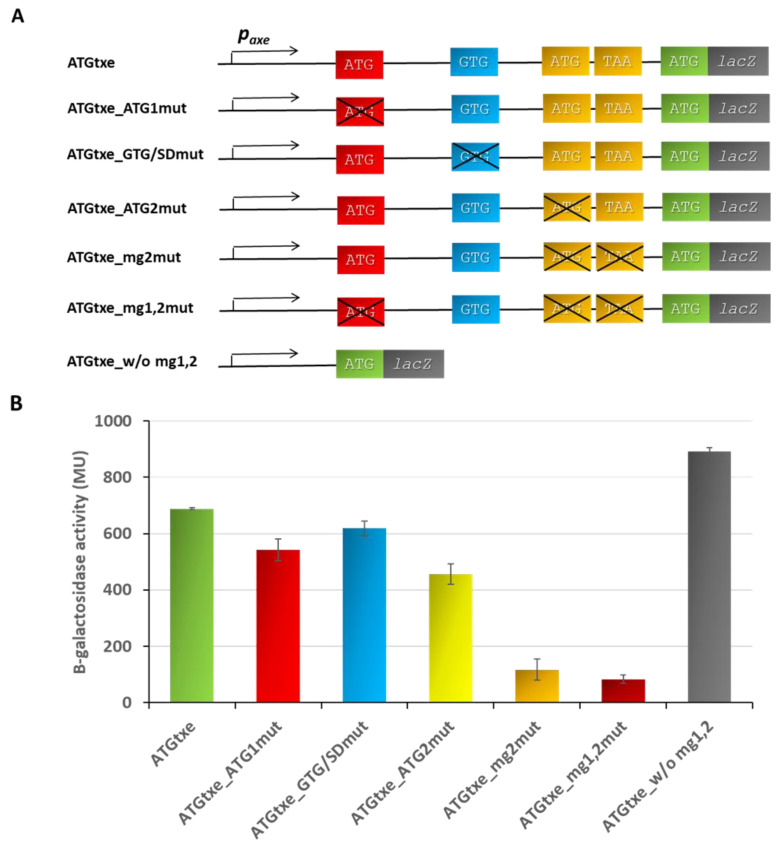
Mutational analysis of the *p_axe_* leader. (**A**) Schematic representation of pRW225_ATGtxe and its derivatives carrying inactivated start and/or stop codons, indicated by black cross. (**B**) β-galactosidase activities were assessed in the *E. coli* ∆5∆lac strain carrying derivatives of pRW225 vector, as indicated. Reporter activity was measured when cell cultures reached OD_600_ ≈ 0.5. These results are the average of at least three independent experiments; error bars represent standard deviation (S.D.). The *p*-values calculated for ATGtxe *versus* all other constructs showed statistical significance (*p* < 0.05). In addition, the difference between ATGtxe_mg2mut *versus* ATGtxe_mg1,2mut is statistically significant.

**Figure 4 ijms-22-13389-f004:**
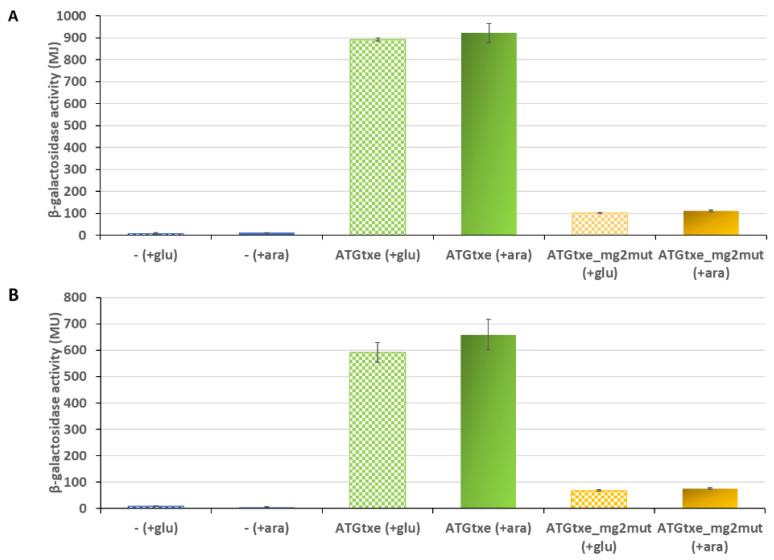
The influence of minigenes supplied *in trans* on reporter protein activity. β-galactosidase activities were assessed in the *E. coli* ∆5∆lac strain carrying empty pRW225 vector (−) as control, or its derivatives, as indicated on the graph, together with pBAD33_mg2 (**A**) or pBAD33_mg1,2 (**B**) bearing DNA fragment containing minigenes under the *p_BAD_* promoter. Expression of the minigenes was induced by the addition of 0.2% L-arabinose (+ara), or repressed by 0.2% glucose (+glu) at the time of inoculation. Reporter activity was measured when cell cultures reached OD600 ≈ 0.5. These results are the average of at least three independent experiments; error bars represent standard deviation (S.D.).

**Figure 5 ijms-22-13389-f005:**
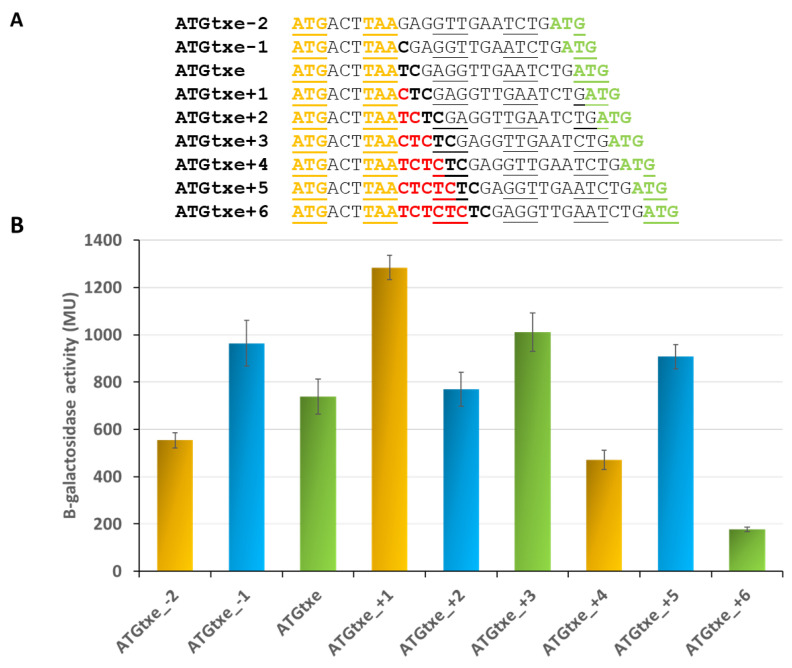
The reading frame and distance between the minigene and the downstream cistron play a role in regulation of *txe* translation. (**A**) Nucleotides that were deleted in two derivatives are indicated in black bold, while inserted nucleotides are displayed in red bold. Codons in the same reading frame as ATG2 (yellow) are underlined. (**B**) β-galactosidase activities were assessed in the *E. coli* ∆5∆lac strain carrying pRW225 derivatives, as indicated on the graph. Reporter activity was measured when cell cultures reached OD600 ≈ 0.5. The same colors are kept for the same open reading frame. These results are the average of at least three independent experiments; error bars represent standard deviation (S.D.). The *p*-values calculated for two constructs in the same reading frames (yellow and green) showed statistical significance (*p* < 0.05), in contrast to those indicated with blue bars where no statistical differences were found between them.

**Figure 6 ijms-22-13389-f006:**
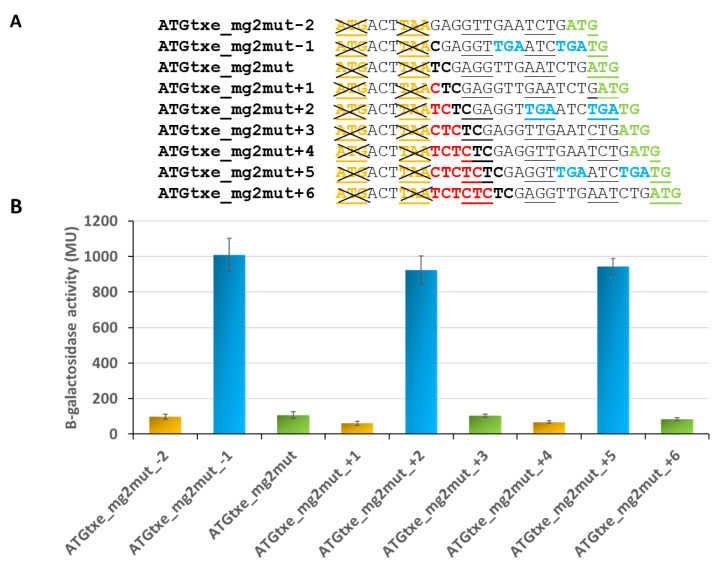
Upon *mg2* inactivation, translation of *lacZ* persists in case of a specific shift in the open reading frame (−1, +2, and +5). (**A**) Nucleotides that were deleted in two derivatives are indicated in black bold, while inserted nucleotides are displayed in red bold. Codons in the same reading frame as ATG2 are underlined. Newly created stop codons are highlighted in bold blue. (**B**) β-galactosidase activities were assessed in the *E. coli* ∆5∆lac strain carrying pRW225 derivatives, as indicated on the graph. Reporter activity was measured when cell cultures reached OD600 ≈ 0.5. The same colors are kept for the same open reading frame. These results are the average of at least three independent experiments; error bars represent standard deviation (S.D.). Calculated *p*-values for blue bars showed no statistically significant differences between them.

**Figure 7 ijms-22-13389-f007:**
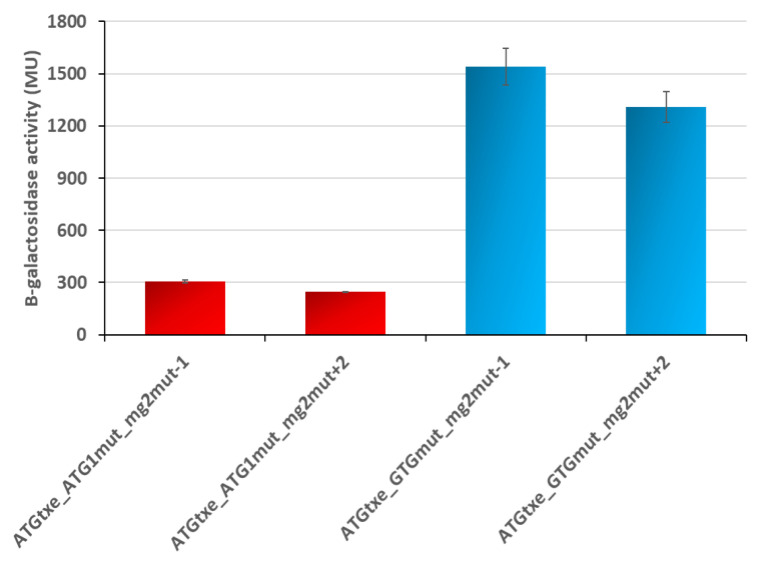
Upon *mg2* inactivation and creation of a new stop codon in the same reading frame, ATG1 takes on a function of a start codon of the new minigene. β-galactosidase activities were assessed in the *E. coli* ∆5∆lac strain carrying pRW225 derivatives, as indicated on the graph. Reporter activity was measured when cell cultures reached OD600 ≈ 0.5. These results are the average of at least three independent experiments; error bars represent standard deviation (S.D.). Calculated *p*-values showed statistically significant differences between ATG1mut and GTGmut derivatives in the corresponding reading frames, −1 and +2, respectively.

**Table 1 ijms-22-13389-t001:** A list of 17 bacterial strains where two-ORF minigenes were identified in their antitoxin DNA sequences. The start codon of a minigene is in the same open reading frame as the start codon of a toxin, with two exceptions indicated by asterisks.

Bacterial Strain	TA Family	Minigene Sequence	Amino Acid ^$^	Number of Nucleotides ^#^
*Agrobacterium tumefaciens* C58 Atu0810	*relBE*	5′-ATGgagTGA-3′	Glu	86 *
*Archaeoglobus fulgidus* DSM 4304 AF2343	*relBE*	5′-ATGagcTGA-3′	Ser	12
*Bartonella henselae* str. Houston-1 BH03400	*relBE*	5′-ATGttgTGA-3′	Leu	147
*Candidatus Protochlamydia amoebophila* UWE25 pc1992	*relBE*	5′-ATGcttTAA-3′	Leu	90
*Enterococcus faecalis* V583 EF0512	*relBE*	5′-ATGattTAG-3′	Ile	174
*Fusobacterium nucleatum subsp. nucleatum* ATCC 25586 FN1099	*relBE*	5′-ATGattTAA-3′	Ile	76 *
*Nitrosomonas europaea* ATCC 19718 NE1562	*relBE*	5′-ATGaatTAG-3′	Asn	57
*Nostoc sp.* PCC 7120 (*Anabaena sp.* PCC 7120) asl2101	*relBE*	5′-ATGactTAA-3′	Thr	108
*Nostoc sp.* PCC 7120 (*Anabaena sp.* PCC 7120) asl2101	*relBE*	5′-ATGccaTAG-3′	Pro	78
*Pectobacterium atrosepticum* SCRI1043 ECA2757	*relBE*	5′-ATGagaTGA-3′	Arg	33
*Salmonella enterica subsp. enterica serovar Paratyphi* A str. ATCC SPA4249	*relBE*	5′-ATGcatTGA-3′	His	57
*Salmonella enterica subsp. enterica serovar Typhi* str. CT18 STY4788	*relBE*	5′-ATGcacTGA-3′	His	57
*Salmonella enterica subsp. enterica serovar Typhi* str. Ty2 t4483	*relBE*	5′-ATGcacTGA-3′	His	57
*Salmonella enterica subsp. enterica serovar Typhimurium* str. LT2 STM4449	*relBE*	5′-ATGcatTGA-3′	His	57
*Streptococcus agalactiae* 2603V/R SAG2009	*relBE*	5′-ATGgttTAA-3′	Val	12
*Nitrosomonas europaea* ATCC 19718 NE1182	*mazEF*	5′-ATGgatTAG-3′	Asp	3
*Shigella flexneri 2a* str. 301 SF3433	*mazEF*	5′-ATGggcTGA-3′	Gly	21

**^#^** The number of nucleotides between the stop codon of the minigene and start codon of the toxin. * Start codon of the minigene is NOT in the same reading frame as the start codon of the toxin. ^$^ Three-letter code for an amino acid encoded in the second position of the minigene.

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
