# Peer review of "Minigene as a Novel Regulatory Element in Toxin-Antitoxin Systems"

_ijms, 2021, doi:10.3390/ijms222413389_

Round 1

Reviewer 1 Report

The type II toxin-antitoxin system is highly regulated to provide tight control of the toxin-antitoxin expression balance. This work reports evidence of a minigene that influences/controls the translation of the Txe toxin. Translational reporter gene fusions were employed to study the molecular basis for this control. The results reveal that two codon mini-ORFs positively affects txe expression by enhanced mRNA stability. It was shown that the cistron reading frame and the distance between them are critical for regulation of the TA system. Sequences of potential minigenes in leader sequences of several toxins in the type II TA family were detected by bioinformatic analysis. These results imply that this gene regulation may be widespread among other TA systems.

General Comments: This study features a comprehensive genetic analysis of the parameters known to influence transcriptional regulation of txe. The results were presented in a logical manner and were methodical in style. This work is an important addition to our understanding of gene regulation in TA systems and strongly recommend that it should be published.

Minor Comments.

Lines 164-166 – protein synthesis was not being directly measured, so this should be re-worded to report a 34% decrease in β-galactosidase activity, not protein synthesis.

Line 173 – “slightly increases this effect." To which effect are you referring? I found this statement confusing.

Lines 379 and 397 – I would advise against using language that suggests txe translation was directly measured, since only changes in the surrogate reporter translation were "indirectly" measured by β-galactosidase activity.

Line 468 – is “melting” the correct word here? Maybe “preventing” or “disrupting”

Language edits (Strikethrough = remove, underline = add)

Line 27 – The toxin-antitoxin (TA) systems

Line 31 – processes

Line 41 – One of such

Line 54 – prompted us here

Line 87 – ORFs

Line 118 – Allowed

Line 341 – in the E. coli cells

Line 354 – In comparison, the construct

Line 376 – that the ribosome’s binding

Line 477 – mechanism by which the minigene

Line 483 – may be (two words)

Author Response

General Comments: This study features a comprehensive genetic analysis of the parameters known to influence transcriptional regulation of txe. The results were presented in a logical manner and were methodical in style. This work is an important addition to our understanding of gene regulation in TA systems and strongly recommend that it should be published.

We thank the Reviewer for positive feedback to our manuscript.

Minor Comments.

Lines 164-166 – protein synthesis was not being directly measured, so this should be re-worded to report a 34% decrease in β-galactosidase activity, not protein synthesis.

This sentence was rephrased according to the Reviewer’s suggestion.

Line 173 – “slightly increases this effect." To which effect are you referring? I found this statement confusing.

We  made correction within the text (lines 167-174) to make this part clearer.

Lines 379 and 397 – I would advise against using language that suggests txe translation was directly measured, since only changes in the surrogate reporter translation were "indirectly" measured by β-galactosidase activity.

Indicated sentences were changed accordingly.

Line 468 – is “melting” the correct word here? Maybe “preventing” or “disrupting”

 According to the Reviewer’s suggestion we replaced the word “melting” by “disrupting”.

Language edits (Strikethrough = remove, underline = add)

Line 27 – The toxin-antitoxin (TA) systems

Line 31 – processes

Line 41 – One of such

Line 54 – prompted us here

Line 87 – ORFs

Line 118 – Allowed

Line 341 – in the E. coli cells

Line 354 – In comparison, the construct

Line 376 – that the ribosome’s binding

Line 477 – mechanism by which the minigene

Line 483 – may be (two words)

We would like to thank the Reviewer for such careful analysis of our manuscript. All language mistakes indicated above were corrected.

Reviewer 2 Report

Manuscript "Minigene as a novel regulatory element in the toxin-antitoxin  systems" by  Barbara Kędzierska and Katarzyna Potrykus, described an interesting regulation of the bacterial toxin-antitoxin system.

The manuscript was written very well and the experiments are clear and support the hypothesis.

The method section needs to elaborate with more detailed experimental procedures.

Specifically, the section of 4.2: cloning and mutagenesis procedures: authors referred one paper, but this section needs details for the construction of every single clone and mutants (insertion, deletion and modifications etc.). These details are very essential for this kind of manuscript and for the researchers to have done this type of study by following the protocol. Please provide those details. 

Author Response

The method section needs to elaborate with more detailed experimental procedures.

Specifically, the section of 4.2: cloning and mutagenesis procedures: authors referred one paper, but this section needs details for the construction of every single clone and mutants (insertion, deletion and modifications etc.). These details are very essential for this kind of manuscript and for the researchers to have done this type of study by following the protocol. Please provide those details. 

According to the Reviewer’s suggestions we added more details in the 4.2 section “Cloning and mutagenesis procedures”. We briefly described different site directed mutagenesis methods which were used in this study and additional citation has been also included – lines 500-521. We indicated which one of these methods was used to make wild type constructs, single nucleotide substitutions, multiple substitutions, as well as deletion and insertion mutants. For each construct relevant information about primers used to create it is presented in Table S1.